# Design, Synthesis, and Biological Evaluation of *N*′-Phenylhydrazides as Potential Antifungal Agents

**DOI:** 10.3390/ijms242015120

**Published:** 2023-10-12

**Authors:** Panpan Zhu, Jinshuo Zheng, Jin Yan, Zhaoxia Li, Xinyi Li, Huiling Geng

**Affiliations:** 1College of Chemistry & Pharmacy, Northwest A & F University, Yangling 712100, China; ppz@nwafu.edu.cn (P.Z.); zjs2021056749@nwafu.edu.cn (J.Z.); yanjin@nwafu.edu.cn (J.Y.); lizhaoxia@nwafu.edu.cn (Z.L.); lxy2022056825@nwafu.edu.cn (X.L.); 2Key Laboratory of Botanical Pesticide R & D in Shaanxi Province, Northwest A & F University, Yangling 712100, China

**Keywords:** *N*′-phenylhydrazide, antifungal activity, structure–activity relationship, antifungal mechanism

## Abstract

Fifty-two kinds of *N*′-phenylhydrazides were successfully designed and synthesized. Their antifungal activity in vitro against five strains of *C. albicans* (*Candida albicans*) was evaluated. All prepared compounds showed varying degrees of antifungal activity against *C. albicans* and their MIC_80_ (the concentration of tested compounds when their inhibition rate was at 80%), TAI (total activity index), and TSI (total susceptibility index) were calculated. The inhibitory activities of 27/52 compounds against fluconazole-resistant fungi *C. albicans* 4395 and 5272 were much better than those of fluconazole. The MIC_80_ values of 14/52 compounds against fluconazole-resistant fungus *C. albicans* 5122 were less than 4 μg/mL, so it was the most sensitive fungus (TSI_B_ = 12.0). **A_11_** showed the best inhibitory activity against *C. albicans* SC5314, 4395, and 5272 (MIC_80_ = 1.9, 4.0, and 3.7 μg/mL). The antifungal activities of **B_14_** and **D_5_** against four strains of fluconazole-resistant fungi were better than those of fluconazole. The TAI values of **A_11_** (2.71), **B_14_** (2.13), and **D_5_** (2.25) are the highest. Further exploration of antifungal mechanisms revealed that the fungus treated with compound **A_11_** produced free radicals and reactive oxygen species, and their mycelium morphology was damaged. In conclusion, the *N*′-phenylhydrazide scaffold showed potential in the development of antifungal lead compounds. Among them, **A_11_**, **B_14_**, and **D_5_** demonstrated particularly promising antifungal activity and held potential as novel antifungal agents.

## 1. Introduction

The World Health Organization has reported the two worst threats to mankind’s health, that is, the continuous increase of antimicrobial resistance for commercial antimicrobials against fungal infections and the decreasing availability of effective therapies [1]. Fungal infections in humans are mainly caused by *Candida*, *Aspergillus*, and *Cryptococcus* [2]. However, *Candida albicans* (*C. albicans*), an adaptable microorganism, can develop resistance to antifungal agents after prolonged exposure to them. It is noteworthy that the emergence of drug-resistant fungi, such as *Candida auris* and *Candida glabrata*, further threatens the limited antifungal drugs [3,4]. As one of the four most common pathogens, *C. albicans* could cause blood infections [5,6]. For instance, the monitoring results of invasive candidiasis in China show that *C. albicans* accounts for 44.9% among a total of 32 *Candida* species [7]. *C. albicans* is a dimorphic fungus that can grow in both yeast and hyphal forms [8]. In infection, its yeast form is more likely to disseminate in the blood, while the hyphal form penetrates tissues through the production of a fungal biofilm, evades phagocytic attack, and adheres to organs [9]. It is estimated that half of adults are infected with *C. albicans* in their oral and gastrointestinal tracts [10]. However, under conditions of a compromised immune system, it can overgrow and cause candidiasis [11]. Currently, inhibiting the transition between the two morphologies with drugs is an important way to reduce the risk of *C. albicans* infection [12]. *Candida* infection has emerged as a significant contributor to mortality among patients with malignant tumors, HIV, and organ transplants, which has resulted in severe consequences for the health of individuals worldwide [13]. It is crucial to address fungal infections in the treatment of patients who have weakened immune systems [14]. According to the literature, developing new antifungal agents and thoroughly understanding the mechanisms of antifungal drug resistance and the use of combination therapy between conventional and repurposed drugs are the current strategies for combating the emerging threat of antifungal resistance in *Candida* [4,15].

Nowadays, various antifungal agents have been developed to cure candidiasis, which can be mainly categorized into four series, including azoles, polyenes, echinocandins, and pyrimidines [16]. Among them, fluconazole (FLC) and caspofungin are the most used clinical drugs [17,18]. However, some strains of fungi have displayed resistance to these drugs, which results in treatment failure [19,20]. Additionally, these antifungal agents may cause side effects such as headache, fever, and liver damage [21], highlighting the urgent need for a new generation of antifungal agents [22].

It is reported that hydrazides exhibit various pharmacological activities, such as inhibiting lipoxygenase, antiviral, antioxidant, and anticancer effects, as well as high inhibitory activity against phytopathogenic fungi and insects. Compounds **1–10** are hydrazides with these biological activities (Figure 1). When both substituents on the sides of hydrazine were phenyl rings, the corresponding compounds **1** and **2** exhibited inhibitory activity against lipoxygenase and adult anopheles gambiae, respectively. Compound **1** displayed a high level of inhibition against lipoxygenase, with a potency of up to 93% observed at a concentration of 10 μM [23]. The LD_50_ value of compound **2** was 220 ng/insect against adult anopheles gambiae [24]. Compounds **3** and **4,** with the carbonyl side substituent of hydrazide being a benzene ring and the hydrazide side substituent being an alkyl or benzyl group, showed antiviral activity and reactivity, respectively. Compound **3** held potent antiviral activity against the histone deacetylase (HDAC) and its IC_50_ values against HDAC1, HDAC2, and HDAC3 were 0.5, 0.1, and 0.06 μM, respectively [25]. Benmoxin (compound **4**) showed modulative reactivity towards Cu (II)–amyloid β and free radicals. Furthermore, its ability to mitigate free radicals was found to be similar to that of trolox, which is a vitamin E analyte [26]. When the carbonyl side substituent of hydrazide is heterocyclic and the hydrazine side substituent is benzene rings or benzyl groups, the corresponding compounds **5** and **6** present inhibitory activity against phytopathogenic fungi and cancer cells. Compound **5** could efficiently inhibit the growth of *Gibberella zeae*, *Fusium oxporum*, *Colletotrichum higgensianum*, and *Score-tiorum*, and their inhibitory rates were 98%, 91%, 100%, and 99%, respectively, at a concentration of 25 μg/mL [27]. Compound **6** could effectively block the T-47D in the G2/M phase of the cell cycle [28]. When the hydrazine side is substituted with a benzyl group and the carbonyl side is substituted with benzyl ether or adamantyl, the corresponding compounds **7**, **8**, and **9** all held antifungal activity against *C. albicans*. Compound **7** demonstrated a minimum inhibitory concentration range of 0.0156–0.125 mg/mL against three strains including *C. albicans*, *C. glabrata*, and *C. tropicalis* [29]. Compounds **8** and **9** also exhibited potent antifungal activity against *C. albicans*, displaying MIC values of 12.5 μM [30]. Isocarboxazid (compound **10**) with benzyl and heterocyclic substituent on the hydrazine and carbonyl side, respectively, had irreversible inhibitory effects on monoamine oxidase [31]. To the best of our knowledge, there is no report about the inhibitory activity of *N*′-phenylhydrazides against *C. albicans*.

In this study, *N*′-phenylhydrazides were designed, synthesized, and structurally characterized. The antifungal activity against five strains of *C. albicans* was assessed via the broth microdilution (BMD) method, and initial structure–activity relationships (SARs) were established. To probe the antifungal mechanism, investigations were carried out involving free radical scavenging, confocal laser scanning microscopy (CLSM), and scanning electron microscopy (SEM).

## 2. Results and Discussion

### 2.1. Design and Synthesis of N′-Phenylhydrazides

The synthetic routes of target compounds are shown in Figure 2. Substituted benzoyl chloride was synthesized by the acylation of substituted benzoic acid with SOCl_2_. And a substitution reaction occurred between it and phenylhydrazine to gain series A compounds. At 0 °C, various substituted phenylhydrazine hydrochlorides were achieved through the diazotization reactions. Then, series B compounds were obtained via the condensation reaction between phenylhydrazine hydrochloride and benzoic acid under EDCI and HOBt. Series C and D compounds were prepared by the condensation reaction of different kinds of carboxylic acids with phenylhydrazine hydrochloride [27,32].

A total of fifty-two kinds of *N*′-phenylhydrazides belonging to four different series were synthesized, with yields up to 96%. The structures of target *N*′-phenylhydrazides were identified by HRMS, ^1^H NMR, and ^13^C NMR spectra. It was discovered that three of them (**D_2_**, **D_4_**, and **D_5_**) were unknown. The separated yields, melting points, and the data of HRMS, ^1^H NMR, and ^13^C NMR spectra of *N*′-phenylhydrazides are illustrated in the Appendix A.

### 2.2. Evaluation of Antifungal Activity In Vitro

The in vitro antifungal activities of target candidates against five strains of *C. albicans* were screened at the concentrations of 0.125, 0.25, 0.5, 1, 2, 4, 8, 16, 32, and 64 μg/mL through the BMD method [33,34]. FLC was selected as the positive control. Based on the data of antifungal activity, their MIC_80_ (the concentration of tested compounds when their inhibition rate was at 80%) and TAI (total activity index) values were calculated according to the formulas MIC_80_ = $10^{\log A+\log\frac{1}{N}\times \frac{a-80}{a-b}}$ and TAI = $\sum_{1}^{n}1/\sqrt{{MIC}_{80}}$, respectively. The TSI value (total susceptibility index) of every strain of fungus was evaluated as well [35]. By analyzing MIC_80_ and TAI values, the preliminary SARs were concluded. According to the standards of the Clinical Laboratory Standards Institute (CLSI), when MIC_80_ values are distributed in ≤4 μg/mL, 8–16 μg/mL, and ≥32 μg/mL, the corresponding compounds are classified as susceptible, intermediate, and resistant candidates, respectively [33]. To evaluate the potency of the compounds with greater inhibitory activity, time–inhibition rate curves were depicted for seven compounds (**A_1_**, **A_2_**, **A_8_**, **A_11_**, **A_12_**, **B_14_**, and **C_8_**) and their respective half-inhibitory times (IT_50_) were calculated.

#### 2.2.1. Antifungal Activity of Target Compounds

A total of fifty-two kinds of compounds were synthesized, their antifungal activities in vitro were screened at a concentration in the range of 0.125–64 μg/mL against five strains of fungi, and their MIC_80_, MFC (minimal fungicidal concentration), and TAI (total activity index) values were calculated (Table 1).

All target compounds displayed definite inhibitory activities against the five tested fungi. In terms of series A compounds, **A_11_** held the best inhibitory activity against *C. albicans* SC5314, 4395, and 5272, and the corresponding MIC_80_ values were 1.9, 4.0, and 3.7μg/mL, respectively. Thus, **A_11_** had the highest TAI value (2.71). **A_8_** (MIC_80_ = 0.7 μg/mL) showed the best inhibitory activity against *C. albicans* 5122. For the *C. albicans* 5172, **A_16_** (MIC_80_ = 5.8 μg/mL) displayed the best inhibitory activity against it.

As far as series B compounds were concerned, **B_3_** (MIC_80_ = 6.9 μg/mL) held the most effective antifungal activity against *C. albicans* SC5314. **B_14_** had the most potent inhibitory activity against *C. albicans* 4395 and 5172 and the corresponding MIC_80_ values were 4.0 and 3.3 μg/mL, respectively. It also had the highest TAI value (2.13). For the *C. albicans* 5122, **B_3_** (MIC_80_ = 0.7 μg/mL) displayed the best inhibitory activity against it. **B_5_** (MIC_80_ = 6.3 μg/mL) showed the best inhibitory activity against *C. albicans* 5272.

As for series C compounds, **C_2_** had the best inhibitory activity against *C. albicans* SC5314 and 5122 and the corresponding MIC_80_ values were 3.4 and 1.7 μg/mL, respectively. For the *C. albicans* 4395 and 5272, **C_4_** displayed the best inhibitory activity against them and the corresponding MIC_80_ values were 8.9 and 3.8 μg/mL, respectively. **C_4_** had the highest TAI value (1.85). **C_9_** (MIC_80_ = 1.4 μg/mL) showed the most powerfully antifungal activity against *C. albicans* 5172.

With respect to series D compounds, **D_2_** exhibited the most potent antifungal activity against *C. albicans* SC5314 and 4395 with MIC_80_ values of 12.2 and 5.5 μg/mL, respectively. As far as the *C. albicans* 5122, 5172, and 5272 were concerned, **D_5_** displayed the best inhibitory activity against them and the corresponding MIC_80_ values were 2.2, 2.7, and 2.4 μg/mL, respectively. Hence, **D_5_** had the highest TAI value (2.25).

#### 2.2.2. The Inhibitory Efficiency of Compounds with Better Antifungal Activity

In order to evaluate the inhibitory potential of the compounds exhibiting better antifungal activity, time–inhibition rate curves were depicted for seven compounds (**A_1_**, **A_2_**, **A_8_**, **A_11_**, **A_12_**, **B_14_**, and **C_8_**) at a concentration of 3.2 μg/mL against *C. albicans* SC5314, and their half-inhibitory times (IT_50_) were calculated (Figure 3 and Figure 4) [33,34].

The time–inhibition rate curve of FLC showed a relatively smooth overall trend, whereas those of the seven tested compounds exhibited a steep rise in the inhibition rate during the 0–10 h period and were followed by a gradual flattening after 20 h. According to calculations, the IT_50_ value of FLC was 14.1 h. In contrast, the IT_50_ values of all seven tested compounds were less than 7.5 h. Compound **A_11_**, in particular, achieved the lowest IT_50_ value (1.4 h). Thus, this class of compounds held a more efficient antifungal efficacy, which might be attributed to their simple structure, making them easy to enter fungal cells.

#### 2.2.3. The Analysis of Preliminary SARs

In this study, the compounds with TAI values ≥ 1.19 were defined as susceptible compounds. When 25% of *N*′-phenylhydrazides had MIC_80_ values ≤ 4 μg/mL against the fungi, the fungi were defined as sensitive. Sixteen susceptible compounds and one sensitive fungus, *C. albicans* 5122, were screened.

According to the above data, the TAI values of twenty-one compounds (**A_2_**, **A_4_**, **A_5_**, **A_6_**, **A_9_**, **A_10_**, **A_11_**, **A_13_**, **A_16_**, **B_1_**, **B_5_**, **B_7_**, **B_10_**, **B_11_**, **B_14_**, **C_1_**, **C_3_**, **C_4_**, **C_7_**, **D_1_**, and **D_5_**) with MIC_80_ ≤ 64 μg/mL were calculated according to the formula TAI = $\sum_{1}^{n}1/\sqrt{{MIC}_{80}}$, which reflected the level of antifungal activity of the compounds, and the larger the index, the better the antifungal activity of the compound [35].

The total susceptibility index (TSI) of the fungi were calculated according to the formula TSI = $\sum_{1}^{n}1/\sqrt{{MIC}_{80}}$, which reflected the sensitivity of every strain of *C*. *albicans*, and the greater the index, the more sensitive the fungus was to these compounds (Table 2) [35].

Standard *C. albicans* SC5314 exhibited a high degree of sensitivity to FLC (TSI = 14.8). For series A and B compounds, the most sensitive fungus was *C. albicans* 5122, and the TSI values were 11.6 and 12.0, respectively. *C. albicans* 5172 showed the highest sensitivity to series C and D compounds, and the TSI values were 7.8 and 11.2, respectively. Thus, *C. albicans* 5122 was a sensitive fungus.

The MIC_80_ and TAI values obtained allowed a preliminary evaluation of SARs, which can be contrasted with **A_1_** to obtain the general trends.

First, the introduction of a chlorine atom to the *ortho*-, *meta*-, or *para*-position of the carbonyl side benzene ring was able to sharply enhance the antifungal activity against all or most of the fungi (**A_5_**, **A_10_**, and **A_15_**). A similar case was observed for *ortho*-/*meta*-F, *para*-Br, and *meta*-/*para*-CF_3_ substituted compounds (**A_2_**, **A_8_**, **A_6_**, **A_11_**, and **A_17_**). The more electron-withdrawing groups on ring A, the better the antifungal activity of the corresponding compounds (**A_12_** > **A_7_**).

Second, the introduction of nine substituents, identical to those present in the series A compounds, onto the phenyl ring of the hydrazine side resulted in compounds with inferior antifungal activity compared to the series A compounds. Among the tested compounds, those with *para*-CH_3_, *para*-Cl, and *para*-F substituents (**B_7_**, **B_11_**, and **B_14_**) exhibited better antifungal activities compared to *ortho*-Cl and *meta*-Br substitutes (**B_1_** and **B_5_**).

Third, the antifungal activity was also influenced by the various heterocycles and alkyl groups. **C_4_** with cyclopropyl significantly improved the antifungal activities toward *C. albicans* 5172 and 5272. However, the activities were significantly cut down by the naphthyl group. A similar case was found for **C_3_**, **C_7_**, and **C_8_** with polar heterocyclic groups (thienyl and furan), which were less effective than **A_1_**. The short-chain alkane counterparts (**C_12_**) possessed better activities against five strains of fungi than their long-chain alkane counterparts (**C_13_**). The longer the carbon chain between the benzene ring and the carbonyl group, the lower the antifungal activity of the corresponding compounds (**C_11_** < **C_10_** < **C_2_**).

Last, combining the pharmacophores of ring A (-F, -Cl, and -CF_3_) and B (-F and -Cl) sharply enhanced the antifungal activity of series D compounds. However, their antifungal activity still did not reach the level of **A_11_**.

### 2.3. The Investigation of the Antifungal Mechanism

In this research, *C. albicans* SC5314 was chosen as the tested fungi, and **A_11_** with the highest TAI (2.71) value was used as the tested compound.

#### 2.3.1. Assay of Free Radical Scavenging

In this assay, glutathione was selected as the free radical scavenger [36]. With an increase in the concentration of glutathione, the inhibition rate of **A_11_** against *C. albicans* SC5314 decreased continuously. When the concentration of glutathione reached 1600 μg/mL, the antifungal effect of **A_11_** essentially disappeared (Figure 5).

At concentrations of 1600, 800, and 400 μg/mL, glutathione exhibited low inhibitory effects on the growth of standard *C. albicans* SC5314, and the inhibition rate of **A_11_** remained consistently stable at around 95%. When the concentrations of glutathione increased, the inhibitory effect of **A_11_** was reduced. In the event that the concentration of glutathione was at 400 μg/mL, the inhibition rate of **A_11_** decreased from 94.7% to 18.4%. On the occasion that the concentration of glutathione was raised to 800–1600 μg/mL, the inhibition rate of **A_11_** dropped to below 20%, similar to the inhibition rate observed when glutathione was used alone. This indicated that **A_11_** produced free radicals during the metabolism process within fungal cells.

#### 2.3.2. Production of ROS

To monitor cellular redox processes, the common oxidative stress indicator DCFH-DA was utilized [35]. As shown in Figure 6, treatment with **A_11_** resulted in a significantly higher fluorescence intensity of DCFH-DA (Figure 6A–C) as compared to the control groups (Figure 6D). These results suggested that **A_11_** could impair mitochondrial function and increase the production of large numbers of ROS in *C. albicans* SC5314.

#### 2.3.3. Effects of **A_11_** on Hyphal Morphology

The impact of **A_11_** on the hyphal morphology against *C. albicans* SC5314 was observed by SEM [37]. In Figure 7A–F, significant morphological changes in the filamentous form of *C. albicans* were observed after treatment with 20 μg/mL of **A11** compared to the control group. The untreated hyphae exhibited dispersed, regular, plump, uniform, and smooth surfaces (Figure 7A,B). In contrast, the treated hyphae appeared shriveled, distorted, with rough surfaces and displayed folding, wrinkling, and invagination (Figure 7C–E). Figure 7F depicts an abnormal transition of the fungus from the yeast form to the filamentous form.

According to the electron microscopy observations, the integrity of mycelia was damaged by **A_11_**. The disruptive effect of **A_11_** on hyphal morphology was the main reason for its inhibitory activity.

#### 2.3.4. Preliminary Antifungal Mechanisms

Based on the research results in Section 2.3.1, Section 2.3.2 and Section 2.3.3, the metabolic processes of **A_11_** within the cell could be preliminarily analyzed. Firstly, **A_11_** was hydrolyzed by fungal amidase to generate phenylhydrazine. In the structure of hydrazine (H_2_N-NH_2_), the lone pairs of electrons on the nitrogen atoms probably repelled each other strongly and led to the reducibility of phenyl hydrazides. Then, under the action of P450, phenyl hydrazine was oxidized to diazo compounds and then released nitrogen gas and generated free radicals. Their reducibility could lead to the reduction of oxygen within cells, forming reactive oxygen species (ROS) as well such as superoxide anions and hydroxyl radicals. Excessive ROS and free radicals disrupted the normal physiological processes of the fungus and subsequently affected its hyphal morphology (Figure 8). This antifungal mechanism was similar to what has been previously reported [38].

The different degrees of inhibitory activity of target compounds against *C. albicans* stemmed from two factors. Firstly, their antifungal activity was affected by the rate of free radical production. The hydrolysis of amide bonds was the first step in the metabolism of *N*′-phenylhydrazines and was also a necessary condition for the formation of free radicals. The electron cloud density of the carbonyl group can be reduced and the hydrolysis of the amide bond can be promoted when the electron-absorbing group is introduced to the substituted benzoic acid, and thus the formation rate of the corresponding free radical is faster. For example, the *ortho*-fluoride, *ortho*-trifluoromethyl, *meta*-chlorine, *meta*-trifluoromethyl, and *para*-cyano substituted compounds (**A_2_**, **A_6_**, **A_10_**, **A_11_**, and **A_16_**) had better antifungal activity compared to the *ortho*-, *meta*-, or *para*-methyl and *para*-methoxy substituted ones (**A_2_**, **A_9_**, **A_13_**, and **A_14_**). Different substituents affect the affinity between the compound and hydrolase, the hydrolysis rate, and the antifungal activity of the compound.

Secondly, their antifungal activity depended on the stability of the generated free radicals, and the poorer the stability of the free radicals, the better the antifungal activity of the compound. The electron-withdrawing group reduced the electron cloud density of the benzene ring, thereby reducing the stability of substituted phenylhydrazine. For example, compounds substituted with *meta-* or *para-* halogen atoms (**B_5_**, **B_11_**, **B_12_**, and **B_14_**) exhibited better antifungal activity compared to compounds substituted with *para* methyl groups (**B_7_**).

The pathogenicity of *C. albicans* is closely associated with the formation of hyphae, which is also an important factor in the formation of biofilms [39]. Typically, the filamentous form of *C. albicans* exhibits greater pathogenicity compared to the yeast form [40]. Under normal growth conditions, the hyphae surface appears uniform, plump, and smooth, without any folds or wrinkles. In the pathological state, the microscopic appearance of the fungal hyphae exhibits signs of invagination and folding [41]. According to Figure 8, it can be observed that the treatment of *C. albicans* with compound **A_11_** resulted in an abnormal morphology of the hyphae. This indicated that compound **A_11_** exhibited a certain degree of inhibition on the formation and growth process of *C. albicans* hyphae. **A_11_** underwent metabolism within the fungal cells and led to the generation of free radicals, and ROS disrupted the normal physiological processes of the fungus and subsequently damaged its hyphal morphology. Next, in vivo antifungal activity and cytotoxicity assay tests will be carried out.

## 3. Materials and Methods

### 3.1. Materials

Substituted benzoic acid, substituted phenyl, substituted phenyl hydrazine hydrochloride, 1-Ethyl-3-(3-dimethylaminopropyl) carbodiimide hydrochloride (EDCI), 1-Hydroxybenzotriazole (HOBt), dried *N*, *N*-Dimethylformamide (DMF), pyridine, heterocyclic carboxylic acids, alkyl carboxylic acid, phenylhydrazine, fluconazole, and dimethyl sulfoxide (DMSO) were all purchased from Adamas Reagent Co. Ltd. (Shanghai, China). Other analytical pure reagents and solvents were bought from local companies. And before use, a number of reagents were purified by standard procedures. Without other explicit direction, water was redistilled and ions were removed before use.

Standard *C. albicans* SC5314 was donated by Professor Dazhi Zhang of the Second Military Medical University. FLC-resistant fungi *C. albicans* 4935, 5122, 5172, and 5272 were donated by Professor Changzhong Wang of Anhui University of Chinese Medicine. At 30 °C, the tested fungi were cultured for 48 h on the reasoner’s 2A (R2A) agar plates. The dilution of suspension was made with RPMI medium 1640; a volume of 1 mL of these suspensions contained 1 × 10^4^–1 × 10^5^ colony forming unit. The tested compounds were diluted at the concentration of 8 μg/mL with DMSO.

A mixture of petroleum ether (PE) and ethyl acetate (EtOAc) was used as the mobile agent and thin-layer chromatography (TLC) used GF_254_ silica gel to monitor the progress of the reactions. An XT-4 micro melting point instrument (Tech Instrument, Beijing, China) was used to measure the melting point of every target compound. An advance neo spectrometer (400 MHz, Bruker, Billerica, MA, USA)) was applied to obtain the ^1^H NMR and ^13^C NMR spectrum of every synthesized chemical. Chemical shifts (*δ* values) and coupling constants (*J* values) were presented in ppm and Hz, respectively. The high-resolution mass spectra (HRMS) was taken advantage of to characterize every unknown compound. On a mass spectrometer (LTQ Orbitrap XL, Thermo Fisher Scientific, Billerica, MA, USA), the positive ion mass spectra of the samples were gained.

### 3.2. The Synthetic Procedure

#### 3.2.1. Synthesis of Substituted Phenylhydrazine Hydrochloride

NaNO_2_ (1.90 g, 27.5 mmol, 1.1 equiv.) was added to a solution of substituted aniline (2.33 g, 25 mmol, 1 equiv.) in hydrochloric acid (60 mL, 20%, *v*/*v*) and stirred at 0 °C for 1 h. Then, the SnCl_2_ (9.48 g, 50 mmol, 2 equiv.) was added and stirred for 2 h at room temperature. The precipitate was filtered and washed with brine and Et_2_O and dried in vacuo at 40 °C overnight [32].

#### 3.2.2. Synthesis of *N*′-Phenylhydrazides

EDCI (8.8 mmol, 1.1 equiv.) and HOBt (8.8 mmol, 1.1 equiv.) were added to a solution of carboxylic acid (8.0 mmol, 1 equiv.) in DMF (20 mL); then phenylhydrazine hydrochloride was added and the reaction mixture was stirred at ambient temperature under a nitrogen atmosphere for 3 h, with TLC detection. The reaction mixture was poured into H_2_O (300 mL), filtrated, and recrystallized by ethanol to give the desired light yellow or brownish-yellow product [27].

### 3.3. Antifungal Activity In Vitro

#### 3.3.1. Determination of MIC_80_ and MFC Value

The antifungal activities in vitro of target compounds against five strains of fungi were determined via the BMD method with assays in 96-well plates as described in the CLSI guidelines (CLSI M27-A3 and M38-A2) [33,34]. FLC was selected as the positive control.

To prepare the stock solutions of the tested compounds, they were dissolved in 100% DMSO with a concentration of 0.8 mg/mL. Before the assay, the solutions were diluted to the concentration of 8.0 μg/mL with RPMI medium 1640. The final concentrations of tested compounds were 64, 32, 16, 8, 4, 2, 1, 0.5, 0.25, and 0.125 μg/mL. After inoculation, the plates were incubated at 30 °C for 24 h. The absorbance of each well was scanned at a wavelength of 625 nm via an ELISA reader (DNM 9602, Perlong, Bingjing, China), and the inhibition rate was calculated.
Inhibition rate (%) = [(A_P_ − A_D_)/(A_P_ − A_N_)] × 100%(1)
where A_P_ is the absorbance of positive control wells, A_D_ is the absorbance of drug wells, and A_N_ is the absorbance of negative control wells.

The MIC_80_ values were the concentration of tested compounds when the inhibition rate was at 80%. A curve was plotted with the logarithm of the concentration as the horizontal axis and the inhibition rate as the vertical axis, and the MIC_80_ was calculated.
(2)$${\mathrm{MIC}}_{80}=10^{\log A+\log\frac{1}{N}\times \frac{a-80}{a-b}}$$

Among them, *A* refers to the corresponding minimum concentration in the aforementioned gradient concentrations when the inhibition rate is just above 80%. *a* is the inhibition rate at concentration *A*, *b* is the inhibition rate just below *a*, and *N* represents the dilution factor.

The MFCs of all target compounds were defined as the lowest concentration of the subculture at which no visible growth was observed on solid media. The assays were carried out in triplicates of three independent experiments.

#### 3.3.2. Time–Inhibition Rate Curves

A quantity of 6.4 mg of the tested compound was dissolved in 0.5 mL of DMSO, then added to RPMI medium 1640 to obtain a drug solution with a concentration of 3.2 μg/mL. The specific procedures are the same as those described in Section 3.3.1. The absorbances were recorded at 0 h, 4 h, 6 h, 10 h, 17.5 h, 22 h, 24 h, 40.5 h, and 70 h, respectively. Every test was conducted three times. The time–inhibition rate curves were plotted by origin software with the time represented on the *x*-axis and the corresponding inhibition rate on the *y*-axis [33,34].

### 3.4. The Investigation of the Antifungal Mechanism

#### 3.4.1. Scavenging of Free Radicals Generated from **A_11_**

The impact of **A_11_** on the emergence of free radicals of *C. albicans* SC5314 was evaluated. Glutathione with gradient concentrations (400, 800, and 1600 μg/mL) was set as a control group to exclude its effect on fungal growth [36]. At the concentration of 20 μg/mL, the inhibition rate of **A_11_** against *C. albicans* SC5314 was close to 100%. μ**A_11_** was added to gradient concentrations (400, 800, and 1600 μg/mL) of glutathione and achieved a concentration of **A_11_** of 20 μg/mL. As shown in Section 3.3.1, the assay should be followed.

#### 3.4.2. Production of ROS

The influence of **A_11_** on the generation of ROS of *C. albicans* SC5314 was ascertained as previously described [42]. DCFH-DA (2′,7′-dichlorodihydrofluorescein diacetate, 40 μM, 0.5 mL) was used to stain the hyphal tips of the fungal mycelium, followed by incubation in the dark at 25 °C for 20 min, and was cleansed with PBS twice, each time for 5 min. At excitation and emission wavelengths of 488 nm and 525 nm, respectively, the fluorescence images were observed and captured before and after stimulation by CLSM (TCS SP8, Leica, Wetzlar, Germany). Images were stored on a computer for further processing and analysis.

#### 3.4.3. Analysis of the Mycelial Morphology

By SEM, the mycelial morphology of *C. albicans* SC5314 was observed according to the reported method [42]. Compound **A_11_** (TAI = 2.71) and standard *C. albicans* SC5314 were chosen as the tested specimen and fungus, respectively.

The fungi were cultured in a thermostatic shaker at 37 °C and 200 rpm for a period of 8 to 24 h. Subsequently, they were soaked in a 4% glutaraldehyde solution for a duration of 4 h. Following this, the fungi were washed three times using a 0.1 mol/L PBS solution with a pH of 7.2. To initiate dehydration, the fungi were exposed to a series of ethanol concentrations including 30%, 50%, 70%, 80%, and 90% (*v/v*). Thereafter, the fungi were treated with 100% ethanol thrice, with each exposure lasting fifteen minutes. Finally, the hyphae were subjected to vacuum drying, coated with gold, and examined through SEM (S-4800, Hitachi, Beijing, China).

## 4. Conclusions

Fifty-two kinds of target compounds were synthesized efficiently and concisely up to 96% yield, and their structures were characterized by HRMS, ^1^H NMR, and ^13^C NMR. It was discovered that three of the compounds (**D_2_**, **D_4_**, and **D_5_**) were unknown.

The target compounds showed different degrees of inhibitory activity against *C. albicans*, among which **A_11_** had the best antifungal activity for all tested fungi, and its TAI value was 2.71, which was higher than that of the positive control FLC (TAI = 1.19). The presence of an electron-absorbing group could significantly enhance the antifungal activity of the compounds in comparison to those having an electron-giving group. Particularly, compound series D showed considerably better antifungal activity than **A_1_**. The IT_50_ values of the seven compounds were about half that of FLC, and **A_11_** was only 1.4 h, which had a more efficient antifungal mechanism.

Further exploration of mechanisms revealed that **A_11_** promoted the generation of endogenous ROS and free radicals, disrupted the physiological processes of *C. albicans* SC5314, and impaired its filamentous morphology, ultimately inhibiting fungal growth.

In summary, **A_11_** is expected to be a novel antifungal agent, and this research holds significant importance for the development of antifungal agents. Next, in vivo antifungal activity and cytotoxicity assays will be carried out.

## Figures and Tables

**Figure 1 ijms-24-15120-f001:**
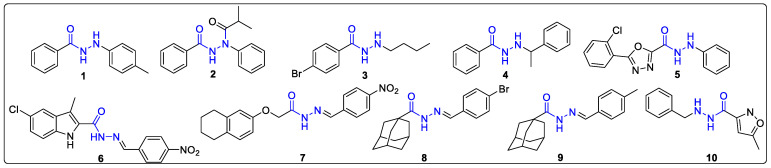
Hydrazides with biological activity.

**Figure 2 ijms-24-15120-f002:**
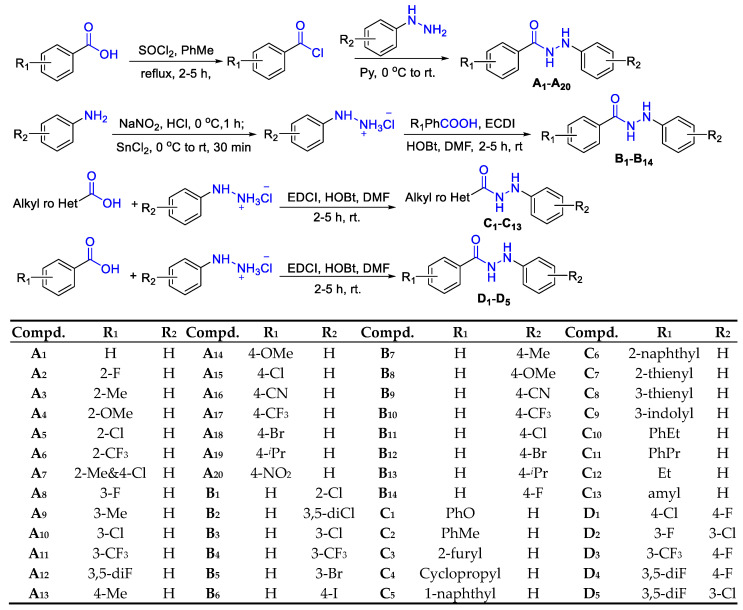
Synthesis of target compounds **A_1_**–**D_5_**.

**Figure 3 ijms-24-15120-f003:**
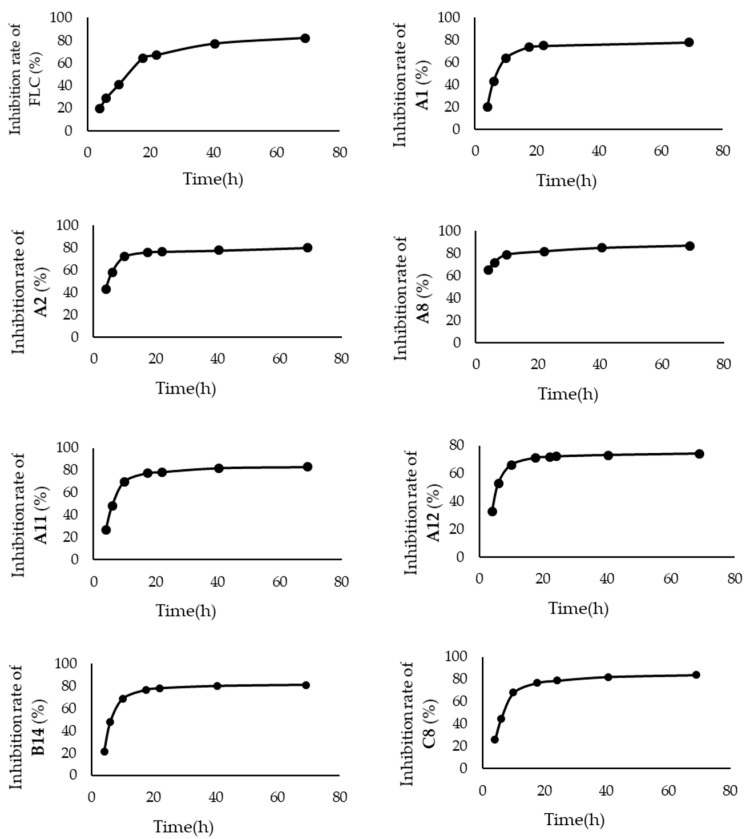
Curves of time–inhibition rate for each compound.

**Figure 4 ijms-24-15120-f004:**
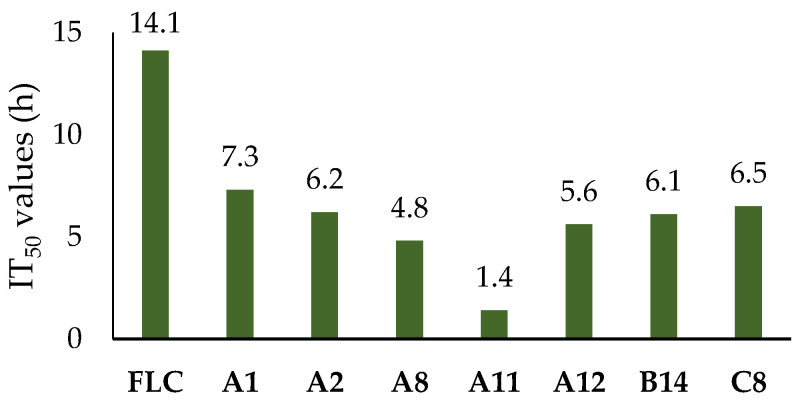
The IT_50_ values of the seven compounds (h).

**Figure 5 ijms-24-15120-f005:**
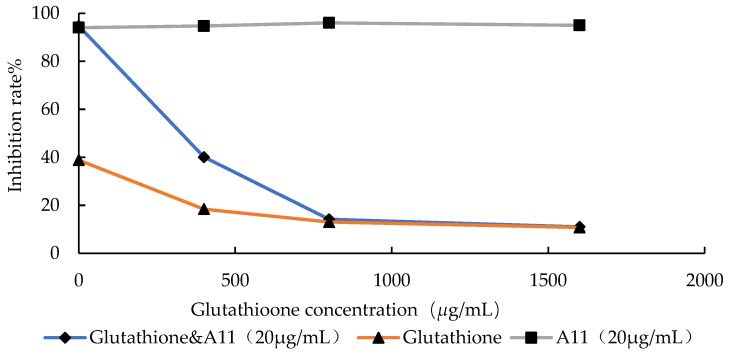
Effect of glutathione on the inhibition rate.

**Figure 6 ijms-24-15120-f006:**
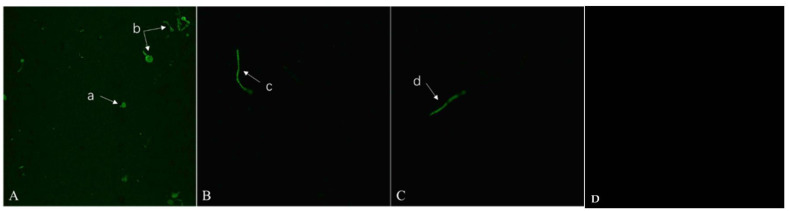
(**A**–**C**), effects of compound **A_11_** on the ROS of *C. albicans* SC5314.; (**D**), control groups. (a), the yeast form of *C. albicans*; (b), the budding state of *C. albicans*; (c, d), the filamentous form of *C. albicans*.

**Figure 7 ijms-24-15120-f007:**
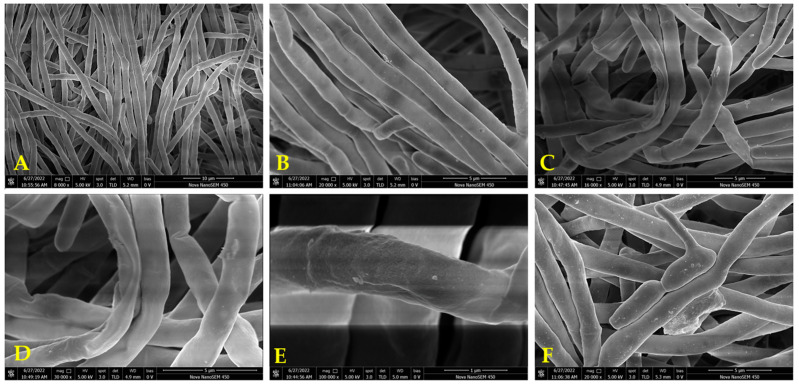
Microstructure of *C. albicans* SC5314 under SEM. (**A**,**B**), untreated hyphae; (**C**–**F**), hyphae treated with **A_11_**.

**Figure 8 ijms-24-15120-f008:**
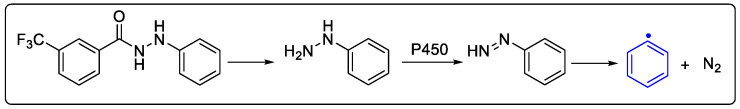
The metabolic process of **A_11_** in fungal cells.

**Table 1 ijms-24-15120-t001:** MIC_80_/MFC values (μg/mL) of target compounds against *C. albicans*.

Compd.	R_1_	R_2_	*C. albicans* SC5314	*C. albicans* 4395	*C. albicans* 5122	*C. albicans* 5172	*C. albicans* 5272	TAI
**A_1_**	H	H	26.0/>64.0	15.0/>64.0	0.9/16.0	>64.0/>64.0	>64.0/>64.0	<1.76
**A_2_**	2-F	H	13.0/64.0	4.1/>64.0	8.5/64.0	6.7/64.0	12.1/>64.0	1.79
**A_3_**	2-Me	H	49.3/>64.0	>64.0/>64.0	13.3/>64.0	21.2/>64.0	53.0/>64.0	N. A.^a^
**A_4_**	2-OMe	H	14.7/>64.0	31.0/>64.0	5.7/>64.0	7.8/>64.0	6.4/>64.0	1.61
**A_5_**	2-Cl	H	25.3/>64.0	52.5/>64.0	10.8/>64.0	13.8/>64.0	13.0/>64.0	1.19
**A_6_**	2-CF_3_	H	11.7/64.0	4.5/>64.0	6.3/64.0	12.0/64.0	22.1/>64.0	1.66
**A_7_**	2-Me&4-Cl	H	26.0/>64.0	>64.0/>64.0	>64.0/>64.0	23.4/>64.0	63.0/>64.0	N. A.
**A_8_**	3-F	H	3.4/64.0	9.7/16.0	0.7/4.0	>64.0/>64.0	27.0/64.0	N. A.
**A_9_**	3-Me	H	25.3/>64.0	61.4/>64.0	10.5/>64.0	14.7/>64.0	17.8/>64.0	1.13
**A_10_**	3-Cl	H	6.9/64.0	6.6/>64.0	15.0/>64.0	12.3/>64.0	59.0/>64.0	1.44
**A_11_**	3-CF_3_	H	1.9/32.0	4.0/>64.0	2.8/64.0	7.4/>64.0	3.7/64.0	2.71
**A_12_**	3,5-diF	H	3.6/>64.0	25.0/>64.0	1.6/8.0	>64.0/>64.0	>64.0/>64.0	N. A.
**A_13_**	4-Me	H	13.9/>64.0	31.0/>64.0	5.0/64.0	14.6/64.0	27.0/64.0	1.35
**A_14_**	4-OMe	H	5.9/>64.0	32.0/>64.0	1.5/8.0	>64.0/>64.0	>64.0/>64.0	N. A.
**A_15_**	4-Cl	H	5.9/>64.0	32.0/>64.0	1.5/>64.0	>64.0/>64.0	>64.0/>64.0	N. A.
**A_16_**	4-CN	H	47.5/>64.0	15.0/>64.0	6.3/>64.0	5.8/>64.0	7.2/>64.0	1.6
**A_17_**	4-CF_3_	H	>64.0/>64.0	15.0/>64.0	4.6/>64.0	9.6/>64.0	9.3/>64.0	N. A.
**A_18_**	4-Br	H	24.4/>64.0	>64.0/>64.0	7.4/>64.0	41.4/>64.0	>64.0/>64.0	N. A.
**A_19_**	4-*^i^*Pr	H	>64.0/>64.0	61.3/>64.0	>64.0/>64.0	>64.0/>64.0	13.5/>64.0	N. A.
**A_20_**	4-NO_2_	H	6.7/>64.0	16.0/>64.0	0.8/4.0	>64.0/>64.0	>64.0/>64.0	N. A.
**B_1_**	H	2-Cl	26.2/>64.0	52.3/>64.0	13.0/32.0	14.0/>64.0	12.5/>64.0	1.16
**B_2_**	H	3,5-diCl	45.3/>64.0	>64.0/>64.0	32.2/>64.0	45.0/>64.0	>64.0/>64.0	N. A.
**B_3_**	H	3-Cl	6.9/>64.0	19.0/>64.0	0.7/4.0	>64.0/>64.0	>64.0/>64.0	N. A.
**B_4_**	H	3-CF_3_	>64.0/>64.0	>64.0/>64.0	31.5/>64.0	58.6/>64.0	>64.0/>64.0	N. A.
**B_5_**	H	3-Br	24.6/>64.0	60.0/>64.0	3.6/64.0	13.9/>64.0	6.3/>64.0	1.52
**B_6_**	H	4-I	16.7/>64.0	>64.0/>64.0	31.8/>64.0	30.0/>64.0	19.0/>64.0	N. A.
**B_7_**	H	4-Me	13.4/>64.0	32.0/>64.0	21.5/>64.0	25.0/>64.0	13.2/>64.0	1.14
**B_8_**	H	4-OMe	9.2/>64.0	22.0/>64.0	0.8/4.0	>64.0/>64.0	>64.0/>64.0	N. A.
**B_9_**	H	4-CN	>64.0/>64.0	>64.0/>64.0	>64.0/>64.0	>64.0/>64.0	62.6/>64.0	N. A.
**B_10_**	H	4-CF_3_	>64.0/>64.0	>64.0/>64.0	>64.0/>64.0	>64.0/>64.0	>64.0/>64.0	N. A.
**B_11_**	H	4-Cl	10.7/>64.0	12.3/>64.0	4.8/>64.0	7.1/>64.0	57.0/>64.0	1.56
**B_12_**	H	4-Br	47.5/>64.0	14.7/>64.0	50.4/>64.0	30.0/>64.0	13.1/>64.0	1.00
**B_13_**	H	4-*^i^*Pr	14.3/>64.0	>64.0/>64.0	21.0/>64.0	52.7/>64.0	>64.0/>64.0	N. A.
**B_14_**	H	4-F	12.2/64.0	4.0/>64.0	3.5/>64.0	3.3/>64.0	14.7/>64.0	2.13
**C_1_**	PhO	H	31.7/>64.0	14.3/>64.0	11.0/>64.0	6.7/>64.0	7.2/>64.0	1.50
**C_2_**	PhMe	H	3.4/16.0	42.0/>64.0	1.7/>64.0	>64.0/>64.0	>64.0/>64.0	N. A.
**C_3_**	2-furyl	H	17.5/>64.0	45.8/>64.0	2.7/>64.0	7.3/>64.0	13.0/>64.0	1.64
**C_4_**	Cyclopropyl	H	58.9/>64.0	8.9/>64.0	7.7/>64.0	3.9/>64.0	3.8/>64.0	1.85
**C_5_**	1-naphthyl	H	>64.0/>64.0	>64.0/>64.0	62.0/>64.0	>64.0/>64.0	>64.0/>64.0	N. A.
**C_6_**	2-naphthyl	H	14.6/>64.0	>64.0/>64.0	51.8/>64.0	58.7/>64.0	>64.0/>64.0	N. A.
**C_7_**	2-thienyl	H	14.2/>64.0	60.0/>64.0	7.0/>64.0	16.5/>64.0	42.0/>64.0	1.17
**C_8_**	3-thienyl	H	13.9/>64.0	22.6/>64.0	11.6/>64.0	14.7/>64.0	>64.0/>64.0	N. A.
**C_9_**	3-indolyl	H	>64.0/>64.0	>64.0/>64.0	>64.0/>64.0	1.4/>64.0	>64.0/>64.0	N. A.
**C_10_**	PhEt	H	51.8/>64.0	16.1/>64.0	15.0/>64.0	>64.0/>64.0	11.0/>64.0	N. A.
**C_11_**	PhPr	H	>64.0/>64.0	>64.0/>64.0	22.8/>64.0	>64.0/>64.0	15.2/>64.0	N. A.
**C_12_**	Et	H	>64.0/>64.0	>64.0/>64.0	13.8/>64.0	>64.0/>64.0	12.1/>64.0	N. A.
**C_13_**	amyl	H	>64.0/>64.0	42.6/>64.0	34.4/>64.0	>64.0/>64.0	28.5/>64.0	N. A.
**D_1_**	4-Cl	4-F	12.2/32.0	5.5/>64.0	6.4/>64.0	3.7/>64.0	29.5/>64.0	1.81
**D_2_**	3-F	3-Cl	>64.0/>64.0	13.8/>64.0	9.6/>64.0	>64.0/>64.0	7.6/>64.0	N. A.
**D_3_**	3-CF_3_	4-F	63.6/>64.0	>64/>64.0	3.8/>64.0	>64.0/>64.0	7.6/>64.0	N. A.
**D_4_**	3,5-diF	4-F	30.4/>64.0	13.5/>64.0	6.3/>64.0	>64.0/>64.0	7.6/>64.0	N. A.
**D_5_**	3,5-diF	3-Cl	64.0/>64.0.	29.3/>64.0	2.2/>64.0	2.7/>64.0	2.4/>64.0	2.25
FLC			1.8/16.0	>128.0/>128.0	58.0/>64.0	>64.0/>64.0	>128.0/>128.0	<1.19 ^b^

^a^: N. A. means that their TAI values are not available; b: due to FLC’s MIC_80_ values being greater than 128 μg/mL for *C. albicans* 4395, the TAI value of FLC was recorded as <1.19.

**Table 2 ijms-24-15120-t002:** TSI values of the specific fungus to all target compounds.

Compd.	SC5314	4395	5122	5172	5272
A	6.4	5.4	11.6	5.8	5.2
B	5.0	4.8	12.0	4.8	5.0
C	5.0	4.2	6.4	7.8	5.8
D	3.4	5.8	9.2	11.2	7.6
FLC	14.8	<1.8 ^a^	2.6	<2.6	<1.8 ^a^

^a^: the MIC_80_ values of FLC for *C. albicans* 4395 and 5272 were greater than 128 μg/mL, so their TSI values were recorded as <1.8.

## Data Availability

Not applicable.

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
