# Peer review of "Design, Synthesis, and Biological Evaluation of N′-Phenylhydrazides as Potential Antifungal Agents"

_ijms, 2023, doi:10.3390/ijms242015120_

Round 1

Reviewer 1 Report

The authors designed, synthesized, and structurally characterized N'-phenylhydrazides. Furthermore, their antifungal activity was evaluated against C. albicans strains. Although the results are promising and has merit, the manuscript needs modification in both scientific information and form. Then, find below my remarks and questions.

Introduction

- I suggest changing the abbreviation “C. alb.” to “C. albicans

- Please rephrase the paragraph in which the hydrazine results are presented. The topics have been inserted without contextualization. For example, what is Compound 1? The authors should briefly explain what these compounds are.

- What is the purpose of Figure 1? There was no discussion of structural similarity between the compounds evaluated and the antifungals.

Results and discussion

- There was a lack of discussion. Have similar compounds been tested by other groups? What were the results? What similarities and differences were found?

- L260 - Additional assays are needed to make this statement. In this case, it's a hypothesis, so I suggest including “probably” or “supposedly”.

- Figure 7 is too small, it's almost impossible to see the details mentioned. I suggest resizing it. 

Materials and Methods

- Which manual was used for the broth microdilution test? CLSI? EUCAST? What number?

- Why was the MIC read at 80%?

- Did the compounds have fungicidal or fungistatic activity?

- What is the origin of the C. albicans strains 4395, 5122, 5172, 5272?

- Please reference all methodologies used.

This information is extremally relevant to the methodology and discussion.

Minor editing of English language required.

Author Response

Dear Professor,

On behalf of all the contributing authors, I would like to express our sincere appreciations of your constructive comments concerning our article. These comments are all valuable and helpful for improving our article.

Q1: I suggest changing the abbreviation “C. alb.” to “C. albicans”.

Reply: We sincerely thank you for careful review. In the revised manuscript, the abbreviation of all “C. alb.” have been substituted with “C. albicans”.

Q2: Please rephrase the paragraph in which the hydrazine results are presented. The topics have been inserted without contextualization. For example, what is Compound 1? The authors should briefly explain what these compounds are.

Reply: According to your scientific advices, in revised manuscript the paragraph in which the hydrazine results are presented has been rephrased, please find it on page 2.

Q3: What is the purpose of Figure 1? There was no discussion of structural similarity between the compounds evaluated and the antifungals.

Reply: The purpose of Figure 1 is to show the structure of commonly used antifungal agents. Actually, we right after discussed the structural similarity between the compounds evaluated and the antifungals in Figure 2, please find it on the page 3.

Q4: There was a lack of discussion. Have similar compounds been tested by other groups? What were the results? What similarities and differences were found?

Reply: Thank you so much for your careful and rigorous review. Hydrazides have been tested by other groups and show that they are rich in biological activity, and we have added these hydrazides for structural differences and similarities. Please find it on the page 2.

Q5: L260-Additional assays are needed to make this statement. In this case, it's a hypothesis, so I suggest including “probably” or “supposedly”.

Reply: I am very grateful for your careful review. According to your suggestion, the word “probably” has been added before “repelled” on line 270, page 9.

Q6: Figure 7 is too small, it's almost impossible to see the details mentioned. I suggest resizing it.

Reply: According to your scientific advices, Figure 7 has been resized and the details mentioned could recognized clearly, please find it on page 9.

Q7: Which manual was used for the broth microdilution test? CLSI? EUCAST? What number?

Reply: We were really sorry for our careless mistake. CLSI was used for the broth microdilution test and the numbers are M27-A3 and M38-A2. In the section of 3.3.1, we have added “The antifungal activities in vitro of target compounds against five stains of fungi were determined via the BMD method with assays in 96-well plates as described in the CLSI guidelines (CLSI M27-A3 and M38-A2)” on line 366-367, please find it on page 11.

Q8: Why was the MIC read at 80%?

Reply: The antifungal activities in vitro of target compounds against five stains of fungi were determined via the microbroth dilution method with assays in 96-well plates as described in the Clinical and Laboratory Standards Institute guidelines (CLSI M27-A3 and M38-A2). According to the protocols M27-A3 from CLSI document, the concentration at which the inhibition rate is higher than 80% to be classified as the minimum inhibitory concentration (MIC80).

Q9: Did the compounds have fungicidal or fungistatic activity?

Reply: The target compounds showed different degrees of inhibitory activity against C. albicans, and their MIC80 and TAI values have been calculated out and list in Table 1, please find it on page 5-6.

Q10: What is the origin of the C. albicans strains 4395, 5122, 5172, 5272?

Reply: FLC-resistant fungi C. albicans 4935, 5122, 5172, and 5272 were donated by Professor Changzhong Wang of Anhui University of Chinese Medicine, please find it on line 336-337, page 11.

Q11: Please reference all methodologies used.

Reply: Thank you so much for your good suggestion. In section 3.3., all methodologies used literatures have been referenced as [32], [34], [35], [36], [41].

Thank you again.

Good luck to you and your research.

Sincerely yours,

Huiling Geng

Professor of Chemistry

Reviewer 2 Report

It is inappropriate to give characterization data for known compounds as if they were novel compounds. It is sufficient to give reference to known compounds. Even if it is desired to be added as supplementary, it should be written in the calculated and found format while writing the HRMS data.

The results obtained by the authors show that they have obtained important antifungal results. The addition of molecular modeling studies, especially of compounds with very good activity, will make the study more attractive.

Almost none of the compounds whose antifungal properties were studied are original. The authors claim that compounds A7, B13, D2, D4, and D5 are novel. But compounds A7 and B13 are not novel. The compounds are available in the corresponding author's recent publication (J. Agric. Food Chem. 2023, 71, 18, 6803–6817).

Above all, although remarkable antifungal results have been obtained, I do not think that reporting the activity results of known compounds is sufficient for publication in this high-quality journal. That's why I recommend you reject the article.

Author Response

Dear Professor,

I am very thankful to you for your kind help and interest. Based on your valuable suggestions, we have revised our manuscript and the details of all changes are listed as below.

Q1: It is inappropriate to give characterization data for known compounds as if they were novel compounds. It is sufficient to give reference to known compounds. Even if it is desired to be added as supplementary, it should be written in the calculated and found format while writing the HRMS data.

Reply: I am very grateful for your kind suggestion. In the revised supporting information, we have written the HRMS data in the calculated and found format. Moreover, we have compared their characterization data of known compounds with reported ones and cited the relevant literatures in supporting information.

Q2: The results obtained by the authors show that they have obtained important antifungal results. The addition of molecular modeling studies, especially of compounds with very good activity, will make the study more attractive.

Reply: Thank you so much for your helpful advice. The research of molecular modeling will take a couple of weeks. At present, we are looking for the target protein or enzyme of A11 and will present the corresponding results in the near future.

Q3: Almost none of the compounds whose antifungal properties were studied are original. The authors claim that compounds A7, B13, D2, D4, and D5 are novel. But compounds A7 and B13 are not novel. The compounds are available in the corresponding author's recent publication (J. Agric. Food Chem. 2023, 71, 18, 6803–6817). Above all, although remarkable antifungal results have been obtained, I do not think that reporting the activity results of known compounds is sufficient for publication in this high-quality journal.

Reply: Thank you for your reminder.

(1) It is reported that hydrazides exhibit various pharmacological activities, such as inhibiting lipoxygenase, antiviral, antioxidant, and anticancer effects, as well as high inhibitory activity against phytopathogenic fungi and insects. To the best of our knowledge, there is no report about the inhibitory activity of N'-phenylhydrazides against C. albicans. Therefore, we systematically explored the antifungal activity of N'-phenylhydrazides and concluded their structure-activity relationships. To our satisfaction, A11, B14, and D5 demonstrated particularly promising antifungal activity and hold potential as novel antifungal agents.

(2) I am very sorry for the careless mistake. In the revised manuscript, compounds A7 and B13 have been listed as known compounds, please find it in line 177 and 424).

(3) In this research, we designed, synthesized, and structurally characterized a series of N'-phenylhydrazides. Furthermore, their antifungal activity was evaluated against C. albicans strains and the results are promising and has merit. To our surprise, three compounds A11, B14, and D5 out of 52 kinds of compounds exhibited potential inhibitory in vitro activity against five strains of C. albicans. And they are considered as promising antifungal agents, particularly A11 and in vivo antifungal activity and cytotoxicity study will be carried out in the future. Consequently, we believe that this manuscript will interest medicinal chemists and organic chemists, and is suitable for publication in first-class journal, the Journal of International Molecular Science.

Thanks again.

Best wishes to you and your research.

Sincerely yours,

Huiling Geng

Professor of Chemistry

Reviewer 3 Report

This paper aims to investigate the discovery of new antifungal agents by the design, synthesis and the bioactivity evaluation of N'-Phenylhydrazides.

This study explained the synthesis of 52 compounds as N’-phenylhydrazide derivatives aiming to provide potent antifungal agents by testing their bioactivity in vitro against five strains of C. alb.

The manuscript is written comprehensively enough to be understandable; the authors addressed this aim by showcasing that proposed mechanism of the antifungal activity for their promising tested compounds.

The paper stated the purpose, discussion and global implication are clearly stated and consistent with the rest of the manuscript; authors provided the required tests and analysis and enough information in their discussion by using a good number of important articles talked about the subject.

The authors addressed their hypothesis and opinion in a reproducible way and proved their results through all the required experiments and analysis and they used enough number of analyses to prove their results. The results were presented in a clear way which facilitate in reaching a conclusion elucidates those three compounds A11, B14, and D5 out of 52 compounds exhibited potential inhibitory in vitro activity against five strains of C. alb. And they are considered as promising antifungal agents, particularly A11 and in vivo antifungal activity and cytotoxicity study will be carried out in the future.

  • To improve the introduction, I suggest that authors should talk more about the antifungal resistance, the following papers might be useful for this purpose:  

1-    Costa-de-Oliveira S, Rodrigues AG. Candida albicans Antifungal Resistance and Tolerance in Bloodstream Infections: The Triad Yeast-Host-Antifungal. Microorganisms. 2020 Jan 22;8(2):154. doi: 10.3390/microorganisms8020154.

2-    Lee Y, Puumala E, Robbins N, Cowen LE. Antifungal Drug Resistance: Molecular Mechanisms in Candida albicans and Beyond. Chem Rev. 2021 Mar 24;121(6):3390-3411. doi: 10.1021/acs.chemrev.0c00199. 

3-    Murphy SE, Bicanic T. Drug Resistance and Novel Therapeutic Approaches in Invasive Candidiasis. Front Cell Infect Microbiol. 2021 Dec 14; 11:759408. doi: 10.3389/fcimb.2021.759408

4-    Lee, Y., Robbins, N. & Cowen, L.E. Molecular mechanisms governing antifungal drug resistance. npj Antimicrob Resist 1, 5 (2023). https://doi.org/10.1038/s44259-023-00007-2

  • Chemical structures (Chemdraw): Authors should re-draw the structure of Caspofungin (Figure 1).

The abbreviations should be explained at the first place they are mentioned.

No plagiarism has been detected.

References: The authors followed the journal guidelines for some references.

No comments

Author Response

Dear Professor,

I am very grateful for your help and attention, we really appreciate all that you have done for our manuscript.

Q1: To improve the introduction, I suggest that authors should talk more about the antifungal resistance, the following papers might be useful for this purpose: 

  1. Costa-de-Oliveira S, Rodrigues AG. Candida albicans Antifungal Resistance and Tolerance in Bloodstream Infections: The Triad Yeast-Host-Antifungal. Microorganisms. 2020 Jan 22;8(2):154. doi: 10.3390/microorganisms8020154.
  2. Lee Y, Puumala E, Robbins N, Cowen LE. Antifungal Drug Resistance: Molecular Mechanisms in Candida albicans and Beyond. Chem Rev. 2021 Mar 24;121(6):3390-3411. doi: 10.1021/acs.chemrev.0c00199.
  3. Murphy SE, Bicanic T. Drug Resistance and Novel Therapeutic Approaches in Invasive Candidiasis. Front Cell Infect Microbiol. 2021 Dec 14; 11:759408. doi: 10.3389/fcimb.2021.759408
  4. Lee, Y., Robbins, N. & Cowen, L.E. Molecular mechanisms governing antifungal drug resistance. npj Antimicrob Resist 1, 5 (2023). https://doi.org/10.1038/s44259-023-00007-2

Reply: We sincerely appreciate the valuable comments. As suggested, we have added more references on antifungal resistance into the INTRODUCTION part in the revised manuscript, please find it on page 1-2.

Q2: Authors should re-draw the structure of Caspofungin (Figure 1).

Reply: Thanks for your good suggestion. We have re-draw the structure of Caspofungin (Figure 1), please find it on page 3.

Q3: The abbreviations should be explained at the first place they are mentioned.

Reply: Thanks for your careful checks. Based on your advice, we have explained the abbreviations of MIC80 at the first place it is mentioned, please find it on line 124, page 3.

Thank you again.

Good luck to you and your research.

Sincerely yours,

Huiling Geng

Professor of Chemistry

Reviewer 4 Report

The study focuses on the synthesis of 52 N’-phenylhydrazides and their evaluation for antifungal activity against various strains of C. albicans. The compounds demonstrated varying levels of activity, with some showing higher inhibitory effects against fluconazole-resistant fungi. Promising compounds such as A11, B14, and D5 exhibited potential as novel antifungal agents. The research highlights the hydrazide scaffold as a promising avenue for antifungal lead compound development.

The paper presents significant findings regarding the antifungal potential of N’-phenylhydrazides. While the study's contributions are commendable, attention is needed to address several grammatical and stylistic issues present throughout the manuscript. These issues may hinder the clarity of the content. Considering a professional language editing service could greatly enhance the manuscript's overall quality and readability. Furthermore, providing the 13C NMR spectra of key compounds like A2, A6, A11, C1, C2, C11, C12, and C13 would offer valuable insights into the structural characterization of the synthesized compounds. This addition would contribute to the thoroughness of the analysis and strengthen the paper's scientific rigor.

This paper needs address several grammatical and stylistic issues.

Author Response

Dear Professor,

We feel great thanks for your professional review work on our article. As you are concerned, there are several problems that need to be addressed. According to your nice suggestions, we have made extensive corrections to our previous draft, the detailed corrections are listed below.

Q1: While the study's contributions are commendable, attention is needed to address several grammatical and stylistic issues present throughout the manuscript. These issues may hinder the clarity of the content. Considering a professional language editing service could greatly enhance the manuscript's overall quality and readability.

Reply: Thanks for your precious suggestion. We have invited a native English-speaking colleague to polish the manuscript. Furthermore, we have tried our best to carefully checked the manuscript and all changes were marked in red color, which will not influence its content and framework.

Q2: Furthermore, providing the 13C NMR spectra of key compounds like A2, A6, A11, C1, C2, C11, C12, and C13 would offer valuable insights into the structural characterization of the synthesized compounds. This addition would contribute to the thoroughness of the analysis and strengthen the paper's scientific rigor.

Reply: I am very sorry to forget supplying the 13C NMR spectra of A2, A6, A11, C1, C2, C11, C12, and C13, please forgive our carelessness. According to your advice, we have supplemented them in the supporting information, please find them on page 17, 21, 26, 51, 52, and 61-63, respectively.

Once again, thank you so much for your comments and suggestions.

Good luck to you and your research.

Sincerely yours,

Huiling Geng

Professor of Chemistry

Round 2

Reviewer 2 Report

Although the authors have made the necessary corrections, my opinion is still the same. I think that this study should not be published in a journal of this quality. As a decision-maker, I care about the editor's opinion. 

Author Response

Dear Professor,

I am very thankful to you for your kind help and comments.

Best wishes to you and your research.

Sincerely yours,

Huiling Geng

Professor of Chemistry